# Motor Control and Regularity of Menstrual Cycle in Ankle and Knee Injuries of Female Basketball Players: A Cohort Study

**DOI:** 10.3390/ijerph192114357

**Published:** 2022-11-02

**Authors:** Elena Vico-Moreno, Andreu Sastre-Munar, Juan Carlos Fernández-Domínguez, Natalia Romero-Franco

**Affiliations:** 1Nursing and Physiotherapy Department, University of the Balearic Islands, E-07122 Palma de Mallorca, Spain; 2Sport High Performance Centre of Balearic Islands, E-07009 Palma de Mallorca, Spain; 3Health Research Institute of the Balearic Islands (IdISBa), E-07120 Palma de Mallorca, Spain

**Keywords:** biomechanical phenomena, menstruation disturbances, lower limb injury

## Abstract

Inadequate motor control facilitates ankle and knee injuries in female basketball. Although biomechanical analysis could help to detect it, aspects such as irregular menstruation make these associations controversial. We aimed to evaluate associations between 2D biomechanics during landing and proprioception with ankle and knee injuries of female basketball players, considering their menstruation regularity. Seventy-one players participated in this study. In the preseason, participants performed a drop-jump to obtain biomechanics during landing and a weight-bearing proprioception test. During the competitive season, all the non-contact ankle and knee injuries were registered. Data showed that 16% of players sustained an ankle or knee injury, being more frequent in players with irregular menstruation compared to regulars (22% vs. 13%, χ^2^ = 6.009, *p* = 0.050, d = 0.6). Players who sustained a left-side injury displayed higher left-side dynamic valgus during landing than uninjured players (χ^2^ = 25.88, *p* = 0.006, d = 1.5). The rest of the variables did not show any significant difference (*p* > 0.05). Monitoring 2D dynamic valgus from a drop-jump could help to detect inadequate motor control that may facilitate ankle or knee injuries of female basketball players, mainly for those with irregular menstruation. Proprioception seems not to be related to injuries.

## 1. Introduction

Ankle and knee injuries have a great impact on female basketball due to the high proportion of these injuries among players and the subsequent social, economic, sports, and psychological consequences [1]. Among reasons, many studies have proposed some characteristics of the female athletes related with altered motor control strategies [2]. Some sports movements during basketball training and competition require landing from a jump or weight-bearing maneuvers that may lead to risky situations and facilitate ankle and knee injuries [3]. Thus, biomechanical evaluation is often considered to determine and classify potential injury risk of players [2]. Since 3D biomechanical analysis of drop jumps requires sophisticated methodologies, the use of a 2D simple method with ordinary cameras has been validated, being also reliable and frequent among health and sport professionals in field conditions [2,4]. Although many studies have evaluated landing strategies in sport population with ankle or knee injuries [5], very few studies have considered this evaluation in a prospective way [6]. Therefore, well-designed prospective studies are needed in order to clarify the role of parameters from 2D biomechanical analysis during landing prior to potential ankle or knee injuries in basketball.

In a similar way, proprioception during weight-bearing tasks has been traditionally used to monitor ankle or knee injuries [7]. Authors often assess proprioceptive errors through repositioning tasks during the injury rehabilitation process as key components to improve [7]. However, to our knowledge, there are no prospective studies to explore the role of lower limb proprioception prior to sustaining injuries. 

Additional to this evaluation, some hormonal parameters such as those related to the menstrual cycle should be taken into account due to their potential capability to affect motor control and facilitate injuries. Previous studies have demonstrated that the hormonal fluctuations that occur during the normal menstrual cycle, mainly between estrogen and progesterone, influence different body systems [8]. It is well-known that estrogen influences muscle contractile properties, muscle repair, regenerative processes and post-exercise muscle damage, with lower values of this hormone being associated with an increased incidence of injury and decreased functional capacity [9]. 

Since female athletes may be more vulnerable to muscle damage (in terms of loss of strength and delayed onset muscle soreness) when sex hormone concentrations are lower, menstrual irregularity should be taken into account when exploring motor control due to the high frequency of this condition among female athletes [10]. High volume and intensity of training, resistance training and energy restriction are frequent factors associated with alterations in reproductive hormones leading to menstrual disruption and/or altered menstruation [11]. 

Based on the aforementioned arguments, it is needed to evaluate motor control strategies before sustaining frequent injuries such as those occurred in the ankle and knee, in order to design specific approaches to prevent it; at the same time, the regularity of menstruation should be considered. This study aimed to evaluate lower limb biomechanics during landing from a drop jump and lower limb proprioception in female basketball players with and without non-contact ankle or knee injuries, prior to suffering them, and considering the regularity of their menstrual cycles. We hypothesized that those injured players would have previously shown worse values of proprioception compared to non-injured players, as well as inadequate strategies during landing in the drop jump. These differences were hypothesized to be higher in female players with irregular menstruation.

## 2. Materials and Methods

### 2.1. Participants

The sample size was calculated through GRANMO application version 7.12 (Spain). Accepting an alpha risk of 0.05 and a beta risk of 0.2 in a two-sided test, 66 participants were needed, with a correlation coefficient of 0.35, and anticipated drop-out rate of 5%. Eighteen female basketball teams (182 basketball players) from the regional female basketball league were contacted, although only 12 agreed to participate (80.1%) (121 female basketball players). As eligibility criteria, participants should be between 17 and 35 years old, have a sports license to play in the regional female basketball league, not have had severe injuries (leading to more than three weeks absence from sport) during the previous six months, and not be using hormonal contraceptives (due to its action to regulate hormonal parameters and decrease injury risk) [12]. Therefore, 78 female basketball players were eligible and voluntarily participated. The follow-up was completed in 71 players. Three players gave up the basketball competition due to personal issues (incompatibility with university education timetable or moving to another city) and four players suffered an injury in other anatomical regions (upper limb or head) (Figure 1). Before participation, all basketball players and/or their legal guardians were informed about the study and signed an informed consent form. The ethics committee of the local university approved this study (Ref. No: 116CER19).

### 2.2. Design and Procedures

A one-season cohort study was conducted by enrolling female basketball players from the regional basketball league. All participants were tested during the 2019–2020 preseason (September 2019) by evaluating biomechanical landing from a drop jump (DJ) test and lower limb proprioception. All the non-contact acute ankle and knee injuries were registered during the 2019–2020 basketball season (September 2019–March 2020). The present study was conducted according to STROBE (STrengthening the Reporting of OBservational studies in Epidemiology) guidelines [13].

During the basketball preseason, body mass and height were assessed in all participants by using a ±100-g precision digital weight scale (Tefal, Rumilly, France) and a t201-t4 adult height scale (Asimed, Barcelona, Spain), respectively. Additionally, all players provided data about their age, basketball experience, leg dominance (leg used to kick a ball), and menstrual regularity cycle (women were asked about their last three menstrual cycles; regular menstruation was considered for those cycles between 24–38 days of frequency, 7–9 days of variation, and 8 days or less of flow; irregular menstruation was considered for any variation of these three parameters) [14]. In the same session, all players performed a lower limb proprioception test and a DJ test. The order to perform both tests was always randomized by using a web-based software (randomizer.org). Before the evaluation, participants completed a 15 min warm-up consisting of 5 min jogging, 5 min ballistic stretching, and 5 min basketball and running exercises. All players performed the baseline assessment at their sports facilities and wore tight-fitting shorts and low-cut athletic shoes. The testing session was supervised by five experienced physiotherapists who instructed all players to perform the warm-up and tests. After warming up and prior to performing the tests, our physiotherapists demonstrated the proprioception test and jump technique. Thus, all players had at least two trials to ensure the familiarization process of both tests. Then, to ensure consistency, a physiotherapist was responsible for attaching reflective markers to each participant in the following anatomical references of both lower limbs: (1) greater trochanter, (2) lateral femoral condyle, (3) fibular neck, (4) lateral malleolus, (5) anterior superior iliac spine, (6) center of the patella, and (7) ankle joint center (middle distance between malleoli, 1 cm distal to lateral malleolus, according to Eng and Winter). Marker locations were based on previous studies [15]. Despite the fact that 3D analysis represents the gold standard research tool, 2D analysis assessment of frontal and sagittal plane joint motion has been demonstrated to be a good alternative to estimate knee joint angles in sport population [16,17]. As advantage, filming screening tests with ordinary video cameras is a low-cost and time-efficient method that may be easily standardized [17]. For this reason, this methodology has been highly employed by sports and health professionals in field conditions [4] and considered valid for many previous studies [16,17].

### 2.3. Measures

Lower limb proprioception: It was evaluated with the joint position sense (JPS) test in the passive–active modality and weight-bearing position, because it requires the motor control of the entire lower limb, as well as by monitoring the knee flexo-extension joint. Both lower limbs were evaluated. The JPS procedure was based on previous studies and performed using the MyProprioception app previously installed on an iPhone X [18]. This iOS application was created by physiotherapists from the University of the Balearic Islands. For the JPS procedure, players were positioned standing with the non-involved lower limb resting on a box, the involved lower limb with a wedge below the heel to rest plantar–flexion musculature, and a chair to avoid losing balance during the JPS test; visual inputs were blocked with a mask, as recommended in previous studies [18]. 

From this initial position, the physiotherapist passively guided participants to a target position (intermediate knee flexion–extension to ensure similar soft tissue stress in all participants) that players should maintain for 5 s before returning to the initial position. Then, players actively reproduced the target position as closely as possible. Each player had three trials (Trial 1, Trial 2, Trial 3). From the JPS test, the following proprioceptive errors were extracted: absolute angular error (AAE) (the absolute difference between the target position and the mean of the three trials, taking no account of the direction of the difference—i.e., the size of the error), relative angular error (RAE) (the arithmetic difference between the target position and the mean of the three trials, taking into account the direction of the difference, which is the same as the trend of error to underestimate or overestimate), and variable angular error (VAE) (the variability of the errors among the three trials) [16].

Drop jump (DJ): The protocol proposed by Padua et al. [19] was applied for the measurement of the DJ vertical jump [19]. All the players were instructed to drop off from a 30-cm high box with their foot positioned shoulder-width distance apart, with their hands on their iliac crests, and upon landing to immediately perform a maximal vertical jump. The jump was not considered valid if players jumped off the box instead of dropping, removed their hands from their iliac crests, lost balance during landing, flexed their knee during the maximal vertical jump, or used one foot to land instead of both feet. In this way, three valid DJs were registered, with a 2 min rest period between trials to avoid fatigue.

Video data with views from the frontal and both left and right sagittal planes were captured to obtain knee joint kinematics by three video cameras capable of recording at 240 frames per seconds (iPad Pro 11, Apple, Cupertino, CA, USA). The camera was positioned at a height of 1.00 m and at length of 1.70 m in both the left and right sagittal planes and the frontal plane. The video analysis was completed using Dartfish ProSuite software (Dartfish Ltd., Fribourg, Switzerland), similarly to previous studies [20].

Measurement of the frontal plane analysis was done using the knee valgus during landing (obtained from knee valgus at initial contact [IC]—peak knee valgus) in both the left and right lower limbs. Measurement of both the left and right sagittal planes analysis was done using knee flexion during landing (obtained from knee flexion at IC—peak knee flexion) in both the left and right legs. IC was considered as the first frame when the player’s feet contacted the floor [21]. For each trial, all measurements were made three times and then averaged. These variables are the most frequently evaluated from the DJ test in previous studies [22]. An illustration of the DJ measurement is provided in Figure 2.

The inter-tester reliability and intra-tester reliability for methods to obtain proprioceptive error from JPS (using MyProprioception app) [18] and kinematic variables from the DJ test (using Dartfish software) were evaluated two weeks before beginning the study using data from five players who voluntarily agreed to perform these tests. Inter-tester reliability was ICC = 0.981 (95% confidence interval—95%CI: 0.966–0.989) for DJ and ICC = 0.943 (95%CI: 0.714–0.989) for JPS (*p* = 0.001), while the intra-tester reliability was ICC = 0.996 (95%CI: 0.992–0.997) for DJ and ICC = 0.948 (95%CI: 0.742–0.990) for DJ (*p* < 0.001).

Injury surveillance system: All the new non-contact acute ankle and knee injuries were registered from September 2019 to March 2020. Non-contact acute injury was considered as any tissue damage derived from sports practice, as a result of a sudden and rapid kinetic energy transfer, without direct or indirect contact with another player or material of the context [23]. To this end, health or sport professionals in each team were contacted weekly by e-mail or phone for the entire season. When an injury was reported, the health professional of the team asked the player to fill out a web-based questionnaire to provide information about the injury mechanism, side, and related symptomatology. Qualified medical doctors ensured medical diagnosis of injuries [6]. Only non-contact mechanisms were considered: during landing, twisting and turning, running, or falling studies [6,24].

### 2.4. Statistical Analysis

Means and standard deviations were obtained for numerical variables (age, height, weight, basketball experience, JPS errors—AAE, RAE, VAE and biomechanics from DJ—knee valgus and knee flexion), and frequencies for categorical variables (menstrual regularity, leg dominance). Normality of data was explored with the Shapiro–Wilk test. Since distribution of data was not normal for any variable, the Kruskal–Wallis (KW) H test was used to compare differences in biomechanical variables between those females who sustained a left-side injury, right-side injury or non-injured players. The variables in this test included left and right knee valgus during landing, left and right knee flexion during landing; left and right absolute, relative, and variable angular errors. To explore pairwise comparisons, the Dunn–Bonferroni post hoc method was used following a significant Kruskal–Wallis test in the non-parametric test procedure. Confidence intervals (CI) 95% were calculated for all differences and Cohen’s d effect sizes (ES) were calculated to determine the magnitude of the differences between groups, interpreted as small (d = 0.2), medium (d = 0.5), large (d = 0.8) or very large (d = 1.3) [25]. The injury incidence in relation of exposure time was obtained by dividing the total number of injuries occurred during the study period by the total exposure time. This result was multiplied by 1000 to obtain the rate of injuries per 1000 h. Injury incidence (%) according to regularity of the menstrual cycle (regular vs. irregular) was evaluated with the Chi-squared test. We used the International Business Machines (IBM) SPSS statistics, version 21.0 (Chicago, IL, USA), and statistical significance was set at *p* < 0.05.

## 3. Results

Table 1 shows anthropometric and socio-demographic characteristics of all the female basketball players.

A total of 13,435 playing hours were registered throughout the 29 week season. Eleven participants (16%) sustained a non-contact ankle or knee injury (five in right-side ankle, two in left-side ankle, three in right-side knee, one in left-side knee) during the season, resulting in an incidence rate of 0.8 injuries per 1000 h of exposure. 

According to the regularity of the menstrual cycle, 22% of irregular players sustained an injury, while 13% of regular players did, with this difference being statistically significant (χ^2^ = 6.009, *p* = 0.050).

Table 2 shows descriptive data from the drop jump and proprioception tests of non-injured, left-side injured or right-side injured female basketball players. When comparing players who remained uninjured with those who sustained a left-side injury and those who sustained a right-side injury, results showed statistically significant differences for left-side knee valgus (*p* = 0.010) and right-side RAE (*p* = 0.030). Post-hoc analysis according to the injured side showed that only left-side injured players displayed higher left-side knee valgus during landing compared to non-injured players, with very large effect size (*p* = 0.006, d = 1.5). No statistically significant difference was found for the rest of the biomechanical and proprioceptive variables (*p* > 0.05).

## 4. Discussion

Our main findings demonstrated that female basketball players who sustained acute non-contact ankle or knee injuries during the basketball season had previously displayed higher dynamic left-side knee valgus during landing from a drop jump compared to uninjured players, these differences being reported for those players who suffered an injury in that lower limb side. In relation to the importance of the regularity of the menstruation of players, a higher proportion of irregular players sustained an injury compared to regular players. No differences were shown regarding to motor control strategies according to the menstruation regularity.

Regarding the analysis of the frontal plane of landing, similar to previous studies [26,27], we observed that the dynamic left-side knee valgus was significantly higher in players who later suffered an ankle or knee injury on that side. It seems that the lack of medio-lateral stabilization shown with knee valgus is a crucial parameter to monitor in order to predict ankle or knee injuries. However, it is needed to highlight that any of these studies evaluated basketball players. 

In reference to the analysis of the sagittal plane of landing, no significant differences were reported in the present study. This finding agrees with most of the previous studies that established >40° as the minimal knee flexion during landing considered appropriate to successfully dissipate energy and mitigate higher mechanical demands for anatomical structures [27]. In our study, knee flexion values were always higher than this cut-off.

We also observed that those female athletes with irregular menstruation were more frequently injured. As we know from previous studies investigating the effects of the menstrual cycle on lower limb kinematics, female athletes have greater knee laxity during the follicular phase, as well as greater knee valgus and tibial external rotation, predisposing factors for joint instability and injury development [28]. Those female athletes with irregular menstrual cycles could be more frequently exposed to this risky situation, since most of the menstrual cycle dysfunctions are due to alterations in estradiol and beta-endorphin, reducing the pulsatility of gonadotropin-releasing hormone and luteinizing hormone. This is a frequent condition of female athletes due to their high level of training, along with other associated typical associated factors such as low energy availability and body composition [11]. 

In reference to weight-bearing lower limb proprioception, our findings showed similar proprioceptive error when comparing those female athletes who latter sustained an ankle or knee injury with those who remained healthy. This similarity was observed for all types of proprioceptive errors (absolute, relative and variable), which could be interpreted that size, direction and variability of the proprioceptive errors, respectively, had no sensitivity to detect potential alterations before injuries. In this regard, the only study to date evaluating the association between repositioning tasks and subsequent lower limb non-contact injuries reported only absolute angular errors [29]. That study evaluated the knee and ankle JPS of 73 U21 high-level male football players during the preseason and recorded all non-contact lower limb injuries that players sustained during the competitive season. As occurred in our investigation, these authors did not find any association between knee or ankle absolute angular error and subsequent lower limb non-contact injuries. 

In the same line, we did not find any significant difference in proprioceptive errors according to the regularity of menstrual cycle. Several studies that have linked lower limb proprioception and the menstrual cycle have concluded that the proprioception of healthy female athletes may vary throughout the menstrual cycle, decreasing their proprioceptive ability when hormone levels are lower, close to menstruation [30,31]. However, other studies suggest that this variation does not occur [32]. Different measurement joints (ankle or knee) and other aspects such as emotional and pain perception could be behind these differences.

Regarding the occurrence of non-contact ankle and knee injuries, our study confirmed the high rate of incidence in female basketball, with 0.8 per 1000 h of exposure and 15.5% of female players injured. These data are similar to previous studies that specifically accounted for non-contact ankle and knee injuries (~13%) [33]. Higher values of injury occurrence that were reported in previous studies could be explained, because we did not account for those injuries sustained during landing on another player’s foot, or similar indirect contact [6,17,34]. In our study, we only considered those new non-contact injuries that were sustained due to mechanisms that may induce high joint and musculoskeletal forces, potentially influenced from a lack of motor control [34]. 

Our study had limitations. Firstly, it is worthy to consider biomechanical results according to the methodology used for drop jump analysis. While 2D biomechanical analysis is a simpler method than 3D analysis, it does not allow one to monitor some key joint motion parameters that occur during the DJ such as knee, hip or trunk rotation, or foot pronation. With this in mind, our results should be specifically considered for 2D methods and not to be extrapolated to those obtained from 3D biomechanical analyses. Although this is an important limitation, we should consider that simple 2D methods are prioritized by practitioners in field conditions due to their easy-to-use and accessible characters. Therefore, using this method may have a direct applicability to field conditions, where other 3D or laboratory instruments are not affordable. Secondly, during our study only 11 ankle and knee injuries were counted, a larger sample size being needed to provide more precise results. Finally, since we recruited several different basketball teams, different training staff members were working with them. Despite this fact, training and match exposure hours were similar among teams. Because the basketball teams recruited belonged to regional female basketball league, our results could not be generalizable to other basketball populations. 

As clinical recommendations, practitioners and other health and sport professionals should be conscious that monitoring biomechanical analysis from landing by using simple 2D methods may be useful to detect biomechanical alterations to be monitored against potential non-contact ankle or knee injuries. There is also a clear need to better understand female physiology and define both the positive and negative effects of hormonal fluctuations during the menstrual cycle in order to prevent injuries and maximize athletic performance.

## 5. Conclusions

Knee valgus during landing from a drop jump, by using 2D simple method, could be a useful parameter to monitor in order to detect inadequate strategies of motor control that may facilitate non-contact ankle and knee injuries in female basketball players. Meanwhile, proprioceptive errors from repositioning weight-bearing tasks did not seem to have the sensitivity to detect biomechanical alterations prior to sustaining injuries. Factors related to hormonal fluctuation derived from menstrual irregularity should not be overlooked as they appear to be predisposing to the development of these injuries. Due to the small number of injuries counted, future studies should continue investigating motor control and proprioception to confirm the role of these motor control parameters prior to non-contact ankle and knee injury occurrence.

## Figures and Tables

**Figure 1 ijerph-19-14357-f001:**
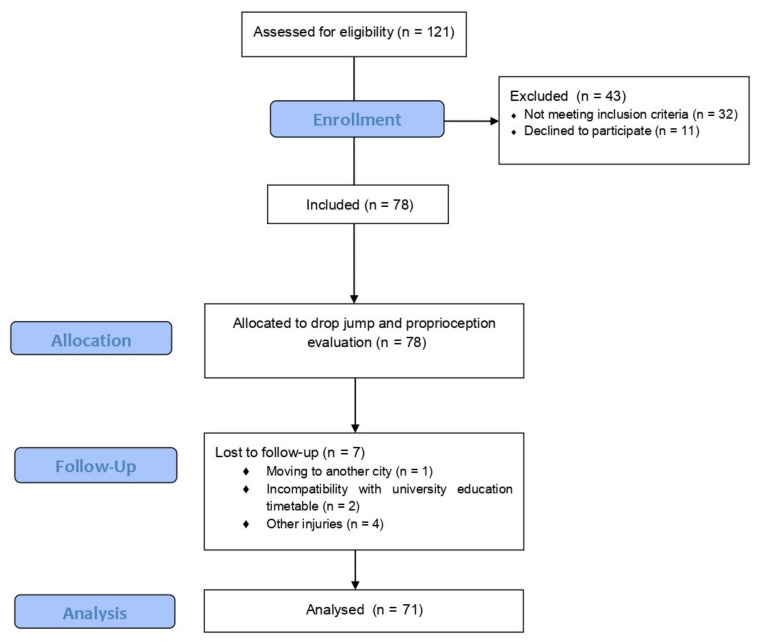
Flow diagram.

**Figure 2 ijerph-19-14357-f002:**
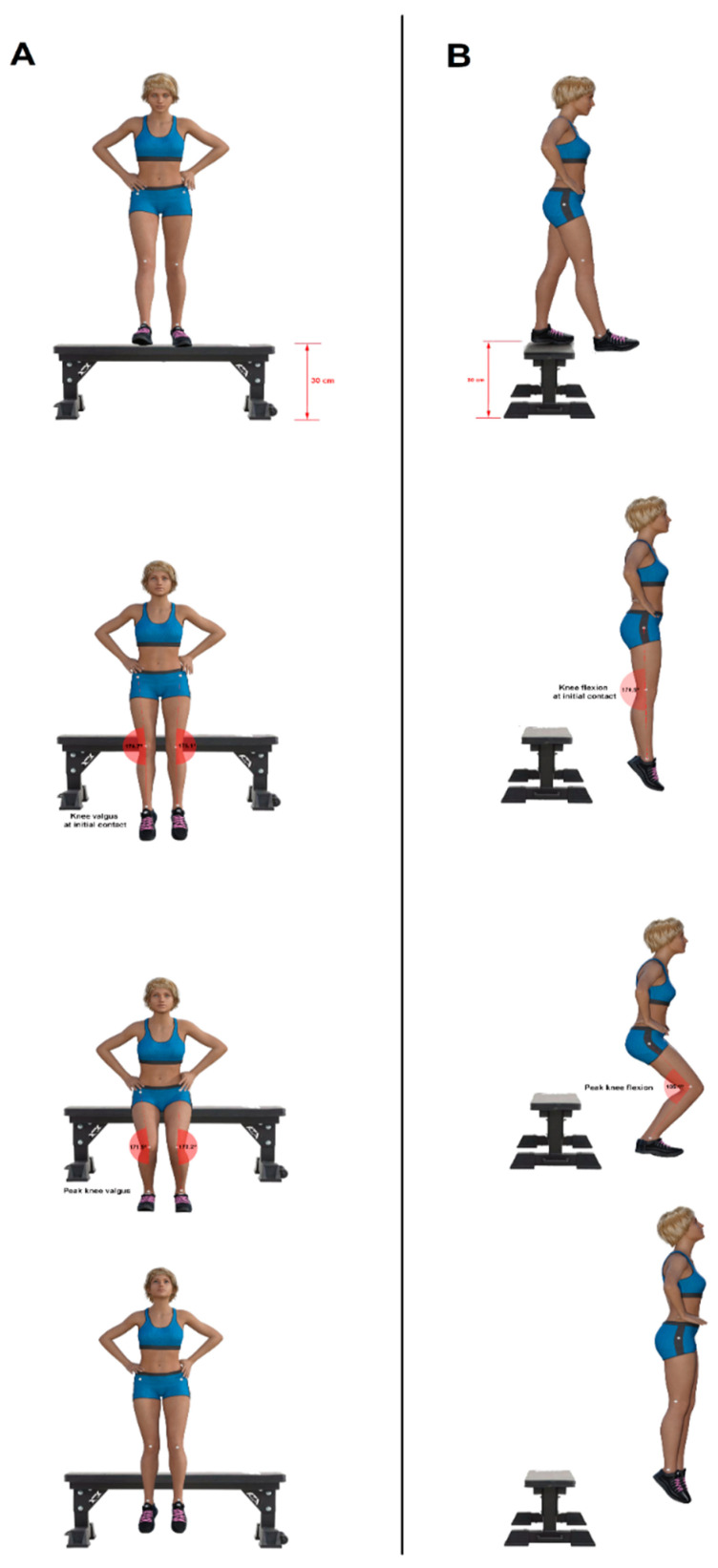
Drop jump measurement. (**A**): frontal plane analysis; (**B**): sagittal plane analysis.

**Table 1 ijerph-19-14357-t001:** Characteristics of female basketball players.

		Players (n = 71)
		Mean ± SD
Age (years)		20.6 ± 5.6
Height (m)		1.70 ± 0.08
Weight (kg)		67.2 ± 13.8
Basketball experience (years)		12.5 ± 5.3
Menstrual cycle	Regular (%)	68
	Irregular (%)	32
Leg dominance	Right (%)	90
	Left (%)	10

SD = standard deviation.

**Table 2 ijerph-19-14357-t002:** Descriptive data for the drop jump and proprioception tests.

		Non-Injured (n = 60)		Right-Side Injured (n = 6)		Left-Side Injured (n = 5)	
Variables (Degrees)		Mean ± SD	95%CI	Mean ± SD	95%CI	Mean ± SD	95%CI
DJ (left-side)	Knee flexion	50.1 ± 13.6	46.5–53.7	55.3 ± 8.9	46.0–64.6	53.8 ± 6.6	45.6–62.0
	Knee valgus	1.9 ± 13.8	−1.7–5.5	−6.6 ± 14.3	−21.5–8.4	−18.8 ± 12.8 **	−34.6–−2.9
DJ (right-side)	Knee flexion	58.5 ± 13.5	55.0–62.0	61.3 ± 11.0	49.8–72.9	56.6 ± 10.3	43.8–69.3
	Knee valgus	−3.1 ± 14.6	−6.8–0.7	−6.2 ± 22.6	−29.8–17.5	−11.7 ± 10.4	−24.6–1.2
JPS (left-side)	AAE	4.1 ± 3.2	3.2–4.9	3.3 ± 3.1	0.1–6.6	4.6 ± 2.1	2.0–7.2
	RAE	2.2 ± 4.6	1.0–3.4	−0.2 ± 4.7	−5.2–4.8	−0.2 ± 5.5	−7.1–6.7
	VAE	1.7 ± 1.2	1.4–2.0	0.8 ± 0.6	0.2–1.5	2.0 ± 1.1	0.7–3.3
JPS (right-side)	AAE	4.4 ± 3.8	3.4–5.4	2.9 ± 2.5	0.3–5.5	4.1 ± 1.3	2.4–5.7
	RAE	0.9 ± 5.7	−0.6–2.4	0.7 ± 4.0	−3.4–4.9	−4.1 ± 1.3	−5.7–−2.4
	VAE	1.6 ± 1.0	1.4–1.9	1.3–0.7	0.6–2.0	1.7 ± 1.2	0.2–3.1

AAE = absolute angular error; CI = confidence interval; DJ = drop jump; JPS = joint position sense; RAE = relative angular error; SD = standard deviation; VAE = variable angular error; Significant differences compared to non-injured players: ** (*p* < 0.01).

## Data Availability

The data presented in this study are available on request from the corresponding author. The data are not publicly available due to privacy.

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
