# Peer review of "Motor Control and Regularity of Menstrual Cycle in Ankle and Knee Injuries of Female Basketball Players: A Cohort Study"

_ijerph, 2022, doi:10.3390/ijerph192114357_

Round 1

Reviewer 1 Report

Dear Authors, The paper seems well conducted in the introduction and partly in the material and methods. The statistical analysis, on the other hand, caused me concern. This is a part that necessarily needs to be clarified and improved! 

Author Response

Dear Reviewer,

Thank you for your helpful comments and your proposal to resubmit our paper entitled: “Motor control and regularity of menstrual cycle in ankle and knee injuries of female basketball players: a cohort study”, pending the correction of these some major issues.

Authors are very grateful for the useful comments and time spent. The modifications have been marked up using the “Track Changes” function such that any changes can be easily located. Also, the replies to reviewer' comments are included, point by point, by explaining the details of the revisions, in bold and red format, in the attached document.

We thank you for your consideration and hope that our responses will come up to your expectations.

Yours sincerely,

The authors

Reviewer 2 Report

The manuscript entitled "Motor control and regularity of menstrual cycle in ankle and knee injuries of female basketball players: a cohort study" presents interesting results and has sufficient quality to be published. However, some minor aspects must be considered. 

Abstract:

You must present the p-value, F, and effect size results in the abstract.

Introduction:

Line 66: use some cites

Material and Methods:

The methodology is correct and complete

Line 83: use a cite

Line 208: use a cite

Results:

Table 1, lines 222, 225: For percentages, use whole numbers without decimals

Line 230: you mention a post hoc analysis. The post hoc test used must be mentioned in the statistical analysis.

Discussion and conclusions:

Is correct and complete, 

Author Response

Dear Reviewer,

Thank you for your helpful comments and your proposal to resubmit our paper entitled: “Motor control and regularity of menstrual cycle in ankle and knee injuries of female basketball players: a cohort study”, pending the correction of these issues.

Authors are very grateful for the useful comments and time spent. The modifications have been marked up using the “Track Changes” function such that any changes can be easily located. Also, the replies to reviewer' comments are included, point by point, by explaining the details of the revisions, in bold and red format, in the attached document.

We thank you for your consideration and hope that our responses will come up to your expectations.

Yours sincerely,

The authors

Round 2

Reviewer 1 Report

The authors clarified my doubts. The paper looks better now. I have no further comments.